

# FairJudge: An Adaptive, Debiased, and Consistent LLM-as-a-Judge

Bo Yang [* 1]   Lanfei Feng [* 2]   Yunkui Chen [2]   Xiao Xu [1]   Yu Zhang [1]   Shijian Li [1]

## Abstract

Existing LLM-as-a-Judge systems suffer from three fundamental limitations: **limited adaptivity** to task and domain-specific evaluation criteria, **systematic biases** driven by non-semantic cues such as position, length, format, and model provenance, and **evaluation inconsistency** that leads to contradictory judgments across different evaluation modes (e.g., pointwise versus pairwise). To address these issues, we propose **FairJudge**, an adaptive, debiased, and consistent LLM-as-a-Judge. Unlike prior approaches that treat the judge as a static evaluator, FairJudge models **judging behavior itself as a learnable and regularized policy**. From a data-centric perspective, we construct a high–information-density judging dataset that explicitly injects supervision signals aligned with evaluation behavior. Building on this dataset, we adopt a curriculum-style SFT–DPO–GRPO training paradigm that progressively aligns rubric adherence, bias mitigation, and cross-mode consistency, while avoiding catastrophic forgetting. Experimental results on multiple internal and public benchmarks show that FairJudge improves agreement and F1 across several evaluation settings, reduces selected non-semantic biases, and achieves competitive or stronger performance than larger general-purpose LLMs on judge-oriented tasks.

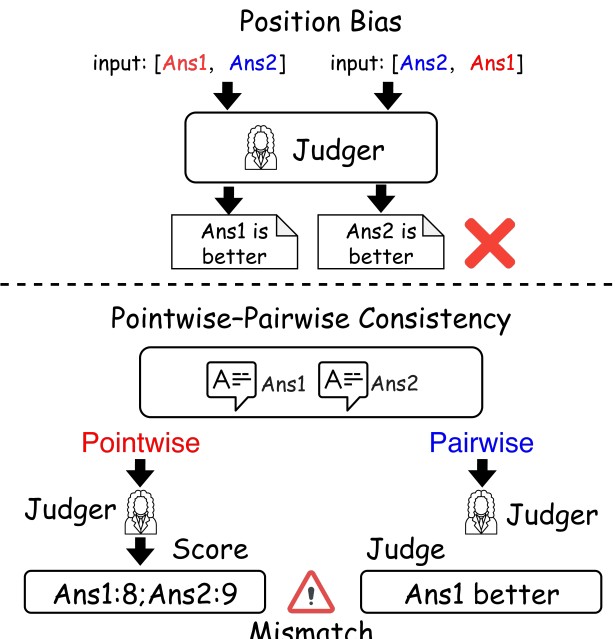

*Figure 1.* Motivation—Two representative issues in LLM-as-a-Judge. Top: **Position bias**, where the judgment flips when the order of answers is swapped. Bottom: **Pointwise–pairwise inconsistency**, where the same answers receive contradictory judgments under different evaluation modes. Representative real-world examples are provided in Appendix Figure 6.

## 1. Introduction

Large language models are increasingly used as automatic judges in model evaluation, preference learning, and supervision pipelines, and have gradually become a critical infrastructure component of modern machine learning systems (Zheng et al., 2023; Chiang et al., 2024). Recent studies have proposed dedicated judge models, datasets, and evaluation benchmarks, demonstrating that targeted training can substantially improve automatic evaluation under specific settings (Zhu et al., 2023; Wang et al., 2023; Christie et al., 2024; Liu et al., 2023; Kocmi & Federmann, 2023). However, as illustrated in Figure 1, in more complex real-world evaluation scenarios, existing LLM-based judges still exhibit notable instability and limited generalization, which constrains their reliability as general-purpose evaluators (Tripathi et al., 2025; Shi et al., 2025; Wang et al., 2024).

These limitations are not merely a consequence of insufficient model capacity or data scale, but reflect a deeper

[*]Equal contribution [1]College of Computer Science, Zhejiang University, Hangzhou, China [2]School of Software Technology, Zhejiang University, Hangzhou, China. Correspondence to: Shijian Li <shijianli@zju.edu.cn>.

*Proceedings of the 43rd International Conference on Machine Learning*, Seoul, South Korea. PMLR 306, 2026. Copyright 2026 by the author(s).

challenge: judging is not a static prediction task, but a behavior decision process jointly constrained by evaluation rules, context, and evaluation settings (Amodei et al., 2016; Hadfield-Menell et al., 2017; Christiano et al., 2017). In practice, evaluation criteria vary across tasks and domains, judgments should be robust to non-semantic perturbations such as answer order, length, or formatting, and evaluation outcomes should remain self-consistent across different evaluation modes (e.g., pointwise versus pairwise) (Tripathi et al., 2025; Shi et al., 2025; Wang et al., 2024). Nevertheless, existing approaches often implicitly assume these properties to emerge naturally, rather than treating them as explicit learning objectives (Ouyang et al., 2022; Bai et al., 2022; Rafailov et al., 2023; Gao et al., 2023).

We argue that the root cause of these issues lies in the fact that judging behavior itself has not been systematically modeled as a learnable and optimizable objective. While prior work has introduced task-specific judge models and datasets, adaptivity, fairness, and consistency in judging are typically assumed rather than jointly constrained during learning (Zhu et al., 2023; Wang et al., 2023; Liu et al., 2023). As a result, LLM-based judges often fail to maintain stable and reliable judgments under rule changes, non-semantic perturbations, or evaluation setting shifts (Tripathi et al., 2025; Wang et al., 2024; Rafailov et al., 2023).

Based on this observation, we propose **FairJudge**, a method that explicitly treats judging behavior as a learning objective, aiming to systematically improve LLM-as-a-Judge systems in terms of adaptivity, debiasing, and consistency. (Gehrmann et al., 2021; Celikyilmaz et al., 2020; Bowman, 2024; Lin, 2004; Papineni et al., 2002; Achiam et al., 2023; Gilardi et al., 2023).

Our main contributions are summarized as follows:

- **FairJudge.** We propose a new perspective that treats judging behavior as a learnable and optimizable objective, and introduce **FairJudge** accordingly, together with the publicly released **FairJudge-16K** training dataset and **FairJudge-Benchmark-1K** evaluation benchmark.

- **Adaptive Judging.** FairJudge explicitly improves the adaptivity of LLM-based judges to task- and domain-specific evaluation criteria by enabling robust alignment with changing rules and contextual requirements.

- **Debiased Judging.** FairJudge systematically mitigates non-semantic biases induced by factors such as answer order, length, and formatting, leading to fairer and more reliable evaluation outcomes.

- **Consistent Judging.** FairJudge significantly enhances judgment consistency across different evaluation settings, including pointwise and pairwise modes, im-

proving the practical reliability of LLM-as-a-Judge systems.

**Conflict of Interest Disclosure.** The authors declare no financial conflicts of interest or other substantive conflicts that could reasonably be perceived to influence the work.

## 2. Related Work

### 2.1. LLM-as-a-Judge and Automated Evaluation

LLM-as-a-Judge has emerged as a practical paradigm for model evaluation and preference-based alignment. Benchmark-driven frameworks such as MT-Bench and Chatbot Arena establish scalable evaluation protocols based on human preferences and pairwise comparisons (Zheng et al., 2023; Chiang et al., 2024). Beyond benchmarks, several works train or adapt LLMs as dedicated judges under fixed evaluation settings, including JudgeLM (Zhu et al., 2023), PandaLM (Wang et al., 2023), and tool-based evaluation systems such as FlexEval (Christie et al., 2024). More recent approaches explore fine-grained or generative judging with explicit rubric conditioning, exemplified by Prometheus (Kim et al., 2023), Generative Judge (Auto-J) (Li et al., 2023), and multilingual judge suites such as M-Prometheus (Pombal et al., 2025). Surveys provide an overview of this rapidly growing area and summarize open challenges in robustness and generalization (Li et al., 2024; Gu et al., 2024).

### 2.2. Adaptive, Debiased, and Consistent Judging

A growing body of work has revealed that LLM-based judges exhibit systematic limitations in adaptivity, debiasing, and consistency. Judgments may be influenced by non-semantic factors such as answer position, length, and formatting, resulting in biased or unfair evaluation outcomes (Wang et al., 2024; Shi et al., 2025; Wang et al., 2025; Zhang et al., 2025; Li et al., 2025; Liu et al., 2024b; Anghel et al., 2025). Length bias, in particular, has been shown to significantly affect automatic preference evaluation, motivating explicit debiasing strategies such as Length-Controlled AlpacaEval (Dubois et al., 2024), as well as additional mitigation analyses (Zhou et al., 2024). Beyond bias, recent studies demonstrate that different evaluation protocols can yield contradictory judgments: pointwise and pairwise evaluation modes may diverge under identical content due to protocol-specific sensitivities (Tripathi et al., 2025). These findings indicate that existing LLM judges often lack robust adaptation to rule changes, effective debiasing against non-semantic cues, and consistent behavior across evaluation settings (Li et al., 2024; Gu et al., 2024).

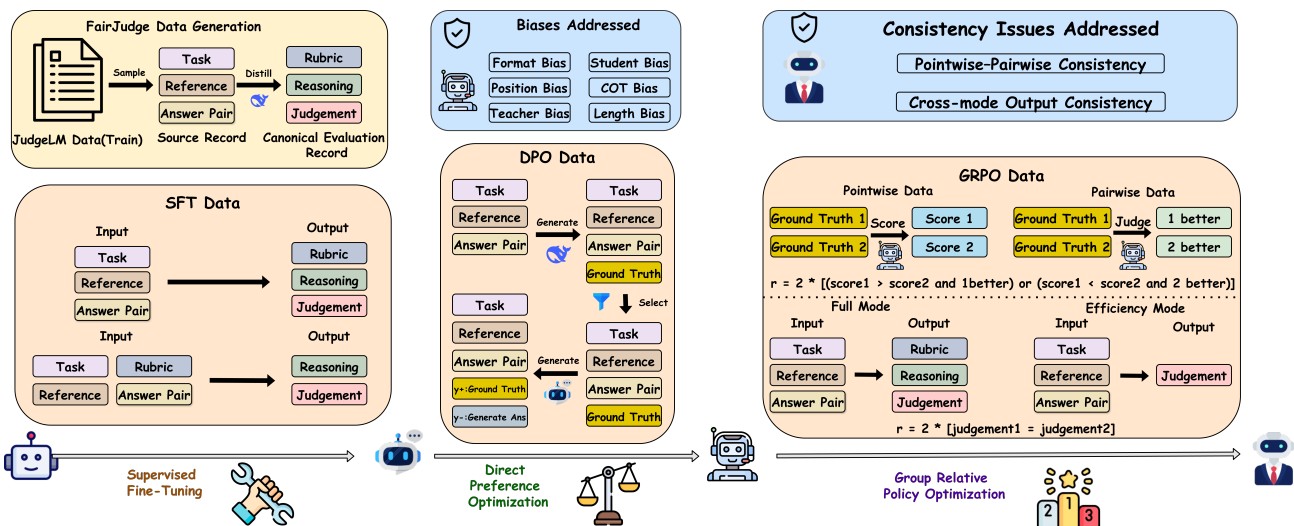

*Figure 2.* Data construction pipeline of FairJudge. **Left (SFT data):** Source evaluation records are organized into {Task, Reference, Answer Pair} and augmented with explicit {Rubric, Reasoning, Judgment}. The rubric is used both as a conditioning input and, in selected cases, as a prediction target. **Middle (DPO data):** Preference pairs (chosen/rejected) are constructed on the same evaluation instance under targeted non-semantic perturbations, guiding the judge to be robust to non-semantic biases. **Right (GRPO data):** Cross-mode samples are organized by jointly constructing **pointwise** (scores or labels) and **pairwise** (relative preference) evaluations for the same instance, with consistency rewards aligning judgments across modes to enforce **cross-mode consistency**.

## 2.3. Preference Learning and Training Judges

From a training perspective, LLM judges are closely related to preference learning and alignment methods. RLHF establishes a foundational paradigm for aligning models with human preferences (Ouyang et al., 2022; Bai et al., 2022; Christiano et al., 2017), while Direct Preference Optimization (DPO) provides a simplified alternative without explicit reward model training (Rafailov et al., 2023). Large-scale AI feedback datasets such as UltraFeedback support scalable preference supervision (Cui et al., 2023), and benchmarks such as RewardBench enable systematic evaluation of reward and judge models (Lambert et al., 2025). At the same time, prior work highlights risks of reward overoptimization and generalization failures, underscoring the need for principled constraints when training judges (Gao et al., 2023).

## 3. Methodology

### 3.1. Problem Formulation: Judging as a Learnable Decision Policy

In most existing LLM-as-a-Judge approaches, the judging process is implicitly modeled as a deterministic mapping from evaluation inputs to output judgments. In the common pointwise setting, a judge directly assigns a score or label to a given input:

$$\text{Judge}_{\text{pt}}(x) \to y,$$

where $x$ denotes the evaluation input (e.g., a question–answer pair) and $y$ represents the final judgment, such as a score, label, or preference. In pairwise evaluation, judging is typically formulated as a comparison function over two candidate answers:

$$\text{Judge}_{\text{pw}}(x_1, x_2) \to y,$$

where $y$ indicates the relative preference between the two inputs.

Under this function-based formulation, evaluation rules and decision logic are implicitly encoded in model parameters or prompts. Differences across tasks, rubrics, or evaluation protocols (e.g., pointwise versus pairwise) are usually handled through mode-specific prompts. As a result, the judging behavior is largely fixed and difficult to control systematically across varying evaluation settings.

In practice, judging behavior depends not only on the input content itself, but also critically on the evaluation context, including task-specific criteria, scoring rubrics, and evaluation modes. The same answer may warrant different yet logically consistent judgments under different rules or protocols. Modeling judging as a single fixed function makes it difficult to capture this conditional dependence and often leads to instability, bias, and inconsistency across settings.

To address this limitation, we explicitly model judging as a *conditional decision policy*:

$$\pi_{\text{judge}}(y \mid x, c, m),$$

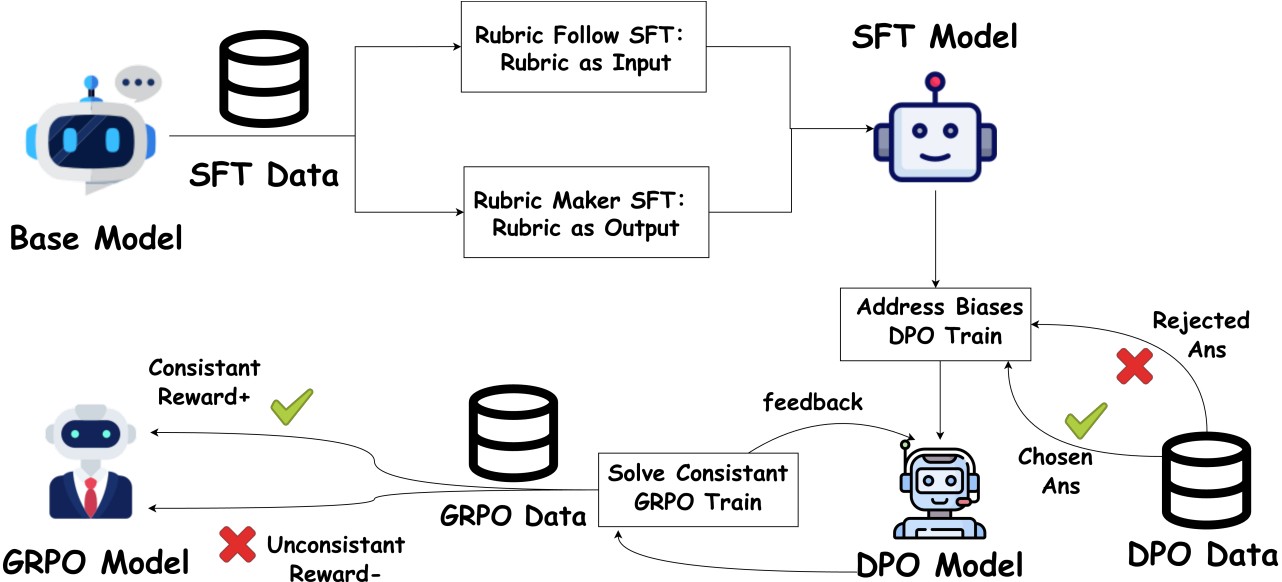

*Figure 3.* Training pipeline of FairJudge. The model is first trained with **SFT data**, where rubrics are used as conditioning inputs and, in some cases, as prediction targets, enabling explicit modeling of evaluation criteria. It is then optimized with **DPO data** in the form of chosen/rejected judgment pairs to reduce non-semantic biases. Finally, **GRPO training** applies consistency-oriented rewards, encouraging judgments that remain stable across evaluation settings. This staged process yields a rubric-aware, debiased, and consistent judge.

where $x$ denotes the evaluation input, $c$ represents the evaluation context (e.g., rubric or reference), $m$ denotes the evaluation mode (such as pointwise or pairwise), and $y$ is the judgment output. Under this formulation, the output $y$ is viewed as a sample from the judging policy conditioned on the evaluation setting, rather than as the unique output of a fixed scoring function.

This policy-based perspective shifts the modeling focus from individual judgment outputs to the overall *behavior* of the judge across conditions. Specifically, the goal of the judging system becomes learning a policy that behaves in a stable, controllable, and generalizable manner under varying contexts and evaluation modes.

In FairJudge, we further characterize the quality of a judging policy along three key dimensions: (1) **Adaptivity**, the ability to adjust judgments in response to different evaluation criteria; (2) **Debiasing**, robustness against non-semantic perturbations such as answer order, length, or formatting; and (3) **Consistency**, the preservation of coherent judgment logic across different evaluation modes. As illustrated in Figure 4, this policy-based formulation provides a principled foundation for subsequent data construction and training, enabling explicit constraints on judging behavior rather than implicit reliance on prompt design. Representative real execution examples of our model are provided in Figure 7.

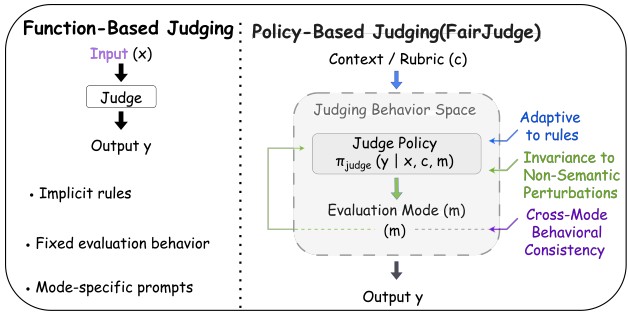

*Figure 4.* **Comparison between function-based judging and policy-based judging.**

## 3.2. Data Construction for Policy-Oriented Judging

To support modeling judging behavior as a learnable and controllable policy, FairJudge does not simply collect preference pairs or scalar scores. Instead, we construct a high–information-density training dataset. Representative examples of the various training data used are provided in Appendix Figure 5.

**Canonicalizing Judging Records.** As shown in Figure 2, FairJudge-16K is constructed from JudgeLM-style evaluation records (Zhu et al., 2023), rather than directly from raw model outputs. Detailed statistics of data distribution and filtering procedures corresponding to Figure 9, Figure 8, 10, and 11 are provided in the Appendix.

*Table 1.* **Comparison of LLM-as-a-Judge performance across three evaluation benchmarks.** The table compares general-purpose multimodal models and specialized judge models on three test sets. PandaLM is a human-annotated benchmark, reflecting alignment with human judgment preferences, while JudgeLM and FairJudge-Benchmark-1K evaluate model robustness and consistency in automated judging scenarios. Baselines include both general LLMs and dedicated LLM-as-a-Judge methods (marked with *). We further report FairJudge results at three model scales (2B / 4B / 8B), all initialized from the corresponding Qwen3-VL backbones, to analyze how judging performance scales with model capacity. The best result in each column is highlighted in **bold**.

| Model | Agreement | Precision | Recall | F1 |
|---|---|---|---|---|
| **PandaLM Test Set (Human Annotations)** | | | | |
| Qwen3-VL-8B | 70.05 | 63.69 | 64.80 | 63.79 |
| GLM-4-9B | 70.02 | 69.86 | 57.69 | 59.55 |
| InternVL3-14B | 66.67 | 65.07 | 68.67 | 62.27 |
| LLaVA-1.5-7B | 37.13 | 40.11 | 39.24 | 28.98 |
| LLaVA-1.5-13B | 29.89 | 43.16 | 41.67 | 28.28 |
| Qwen2.5-72B | 72.80 | 66.25 | 63.60 | 64.58 |
| DeepSeek-V3-671B | 64.76 | 58.48 | 61.13 | 58.83 |
| FlexVL-7B* | 71.34 | 64.30 | 62.18 | 63.00 |
| JudgeLM-7B* | 66.77 | 63.83 | 71.95 | 63.92 |
| PandaLM-7B* | 65.82 | 44.56 | 48.93 | 46.24 |
| FairJudge-2B | 71.53 | 67.75 | 59.99 | 61.68 |
| FairJudge-4B | 69.31 | 64.65 | 61.02 | 62.18 |
| FairJudge-8B | **76.83** | **71.87** | **72.54** | **72.18** |
| **JudgeLM Test Set** | | | | |
| Qwen3-VL-8B | 72.15 | 58.77 | 58.78 | 58.72 |
| GLM-4-9B | 72.42 | 59.06 | 53.31 | 52.54 |
| LLaVA-1.5-7B | 40.80 | 43.11 | 41.16 | 34.34 |
| LLaVA-1.5-13B | 48.59 | 41.61 | 40.45 | 39.69 |
| InternVL3-14B | 71.96 | 62.26 | 64.69 | 61.23 |
| Qwen2.5-72B | **79.59** | **71.31** | 60.95 | 61.92 |
| DeepSeek-V3-671B | 72.04 | 59.38 | 60.37 | 59.75 |
| FlexVL-7B* | 77.44 | 64.82 | 57.49 | 57.42 |
| JudgeLM-7B* | 78.00 | 66.02 | **67.61** | 66.66 |
| PandaLM-7B* | 67.42 | 44.99 | 48.53 | 46.66 |
| FairJudge-2B | 71.20 | 59.88 | 52.36 | 51.48 |
| FairJudge-4B | 75.07 | 71.03 | 59.59 | 60.95 |
| FairJudge-8B | 78.82 | 66.77 | 66.93 | **66.78** |
| **FairJudge-Benchmark-1K** | | | | |
| Qwen3-VL-8B | 63.93 | 62.85 | 56.45 | 58.35 |
| GLM-4-9B | 64.40 | 60.79 | 56.29 | 57.75 |
| InternVL3-14B | 58.31 | 57.76 | 62.30 | 55.11 |
| LLaVA-1.5-7B | 40.34 | 37.42 | 35.49 | 32.57 |
| LLaVA-1.5-13B | 45.86 | 38.93 | 41.95 | 39.03 |
| Qwen2.5-72B | 67.45 | 69.23 | 57.39 | 59.72 |
| DeepSeek-V3-671B | 55.97 | 49.97 | 51.66 | 50.26 |
| FlexVL-7B* | 59.58 | 49.56 | 46.01 | 44.72 |
| JudgeLM-7B* | 69.56 | 65.27 | **67.41** | 66.11 |
| PandaLM-7B* | 52.46 | 35.00 | 39.23 | 36.99 |
| FairJudge-2B | 62.17 | 58.89 | 58.15 | 56.37 |
| FairJudge-4B | 67.74 | **72.19** | 59.97 | 62.57 |
| FairJudge-8B | **71.50** | 71.15 | 66.92 | **67.63** |

*Table 2.* **Ablation Study of FairJudge.** We evaluate the effect of removing different training stages (SFT, DPO, GRPO) on three test sets.

| Model | Agreement | Precision | Recall | F1 |
|---|---|---|---|---|
| **PandaLM Test Set (Human Annotations)** | | | | |
| BASE(Qwen3-8B) | 69.63 | 54.31 | 59.84 | 54.05 |
| W/O SFT | 74.07 | 67.85 | 70.42 | 68.74 |
| W/O DPO | 75.70 | 70.59 | 69.91 | 70.05 |
| W/O GRPO | 74.50 | 68.53 | 71.21 | 69.45 |
| FairJudge-8B | **76.83** | **71.87** | **72.54** | **72.18** |
| **JudgeLM Test Set** | | | | |
| BASE | 73.84 | 59.89 | 60.51 | 59.93 |
| W/O SFT | 76.32 | 63.80 | 64.14 | 63.86 |
| W/O DPO | 77.58 | 65.56 | **67.17** | 66.11 |
| W/O GRPO | 76.78 | 64.13 | 64.02 | 63.95 |
| FairJudge-8B | **78.82** | **66.77** | 66.93 | **66.78** |
| **FairJudge-Benchmark-1K** | | | | |
| BASE | 66.35 | 59.65 | 59.54 | 59.54 |
| W/O SFT | 68.62 | 71.00 | 56.12 | 57.86 |
| W/O DPO | 68.15 | 64.12 | 58.13 | 59.64 |
| W/O GRPO | 68.85 | **72.32** | 57.39 | 59.69 |
| FairJudge-8B | **71.50** | 71.15 | **66.92** | **67.63** |

Each source record is restructured into a canonical evaluation instance consisting of a *task*, *reference information*, *answer pair*, *rubric*, *reasoning*, and *final judgment*. This canonicalization explicitly exposes the contextual conditions under which judgments are made, transforming previously implicit evaluation criteria into learnable signals.

**Structured Construction for Debiasing.** To address non-semantic biases commonly observed in LLM-based judges, FairJudge-16K introduces systematic contrastive constructions at the data level (middle of Figure 2). While preserving semantic equivalence, evaluation inputs are perturbed along non-semantic dimensions such as answer order, length, formatting, reasoning style, and teacher model provenance. These structured perturbations explicitly constrain the judging policy to be invariant to non-semantic factors, allowing debiasing to be learned as a behavioral property rather than enforced through prompt heuristics.

**Cross-Mode Consistency Supervision.** Beyond bias mitigation, FairJudge-16K explicitly targets consistency across evaluation modes. As illustrated on the right side of Figure 2, the same evaluation content is paired with both pointwise and pairwise supervision, enabling direct alignment between different judgment formats. This cross-mode pairing provides the necessary foundation for consistency-oriented optimization in later training stages, preventing contradictory judgments across evaluation protocols.

## 3.3. FairJudge-Benchmark-1K

During the construction of the FairJudge training data, we simultaneously curated and froze a scale-controlled evaluation subset, referred to as **FairJudge-Benchmark-1K**. Rather than being created post hoc after model training, this benchmark naturally emerged as a by-product of the data generation pipeline, and is designed to characterize judge behavior under realistic evaluation conditions.

Specifically, FairJudge-Benchmark-1K is independently sampled from the normalized evaluation records, covering diverse task types, evaluation rubrics, and evaluation modes (e.g., pointwise and pairwise). Each instance preserves the full evaluation context, candidate answer pairs, and their canonicalized judgments, enabling systematic analysis of adaptivity, debiasing, and cross-mode consistency in LLM-based judges.

To ensure evaluation reliability and strict separation from training, we incorporate a **human-in-the-loop** auditing process during the construction and freezing of this subset. Human inspection is limited to verifying structural integrity, consistency of evaluation metadata, and potential data leakage risks, without re-annotating or subjectively modifying judgment outcomes. Once verified, the benchmark is fully frozen and excluded from all stages of model training, preference optimization, and policy learning.

Since FairJudge-Benchmark-1K shares the same unified data generation and normalization pipeline as the training corpus, it remains distributionally aligned in terms of task structure and evaluation format, while being strictly disjoint at the instance level. This design allows the benchmark to reliably reflect judge performance in practical evaluation scenarios without introducing additional assumptions or confounding factors.

## 3.4. Training Paradigm: Curriculum Optimization of Judging Policies

As illustrated in Figure 3, FairJudge is trained through a three-stage curriculum that progressively aligns the judging policy with explicit evaluation rules, debiasing constraints, and cross-mode consistency requirements. Rather than treating training as a generic SFT–DPO–RL pipeline, each stage in FairJudge introduces a distinct and necessary supervision signal that targets a specific property of judging behavior.

**Stage I: Supervised Fine-Tuning for Rubric-Aware Judging.** The first stage aims to establish a stable and controllable judging behavior space. We perform supervised fine-tuning (SFT) using high-quality canonical evaluation records. Each training instance includes the evaluation task, reference context, candidate answer(s), and an explicit rubric. The model is trained to generate structured judgments that explicitly condition on the provided rubric and context.

This stage serves two purposes. First, it enforces rubric-following behavior by making evaluation criteria an explicit part of the model input. Second, it stabilizes the output format (e.g., reasoning followed by judgment), which is essential for subsequent preference-based optimization. At this stage, the judge learns *how to judge*, but not yet *how to judge robustly*.

**Stage II: Debiasing via Direct Preference Optimization.** While SFT aligns the judge with evaluation rules, it does not prevent the model from exploiting non-semantic cues such as answer length, position, or formatting. To mitigate such biases, we introduce a debiasing stage based on Direct Preference Optimization (DPO). Details about DPO are provided in Appendix A.1.

We construct preference pairs where the *chosen* judgment adheres to the rubric while remaining invariant to non-semantic variations, and the *rejected* judgment exhibits systematic bias. DPO then optimizes the judging policy to prefer unbiased evaluation behaviors without requiring an explicit reward model. This stage reduces sensitivity to spurious cues while preserving rubric adherence learned during SFT.

**Stage III: Consistency Optimization via Group-Relative Policy Optimization.** The final stage addresses a critical but underexplored challenge: maintaining consistent judgments across different evaluation modes, such as pointwise and pairwise settings. Instead of relying on prompt engineering or post-hoc calibration, FairJudge explicitly encodes cross-mode consistency as an optimization objective.

We adopt Group-Relative Policy Optimization (GRPO), where multiple judgments are sampled for the same evaluation instance under different modes. A consistency-aware reward assigns higher scores to judgment groups that yield logically equivalent outcomes across modes, and penalizes contradictory decisions. By optimizing group-relative advantages, GRPO encourages stable and self-consistent judging behavior while avoiding reward overfitting. Details about GRPO objective and GRPO reward are provided in Appendix A.2 and Appendix A.3.

**Overall Objective.** The full training objective combines the three stages in a curriculum manner:

$$\mathcal{L} = \mathcal{L}_{\text{SFT}} + \lambda_{\text{DPO}}\mathcal{L}_{\text{DPO}} + \lambda_{\text{GRPO}}\mathcal{L}_{\text{GRPO}}, \qquad (1)$$

where each component corresponds to a distinct behavioral constraint. Together, this curriculum enables FairJudge to learn a judging policy that is adaptive to evaluation rules, robust to non-semantic biases, and consistent across evaluation protocols.

*Table 3.* **Multimodal Benchmarks (MLLM-as-a-Judge, Human Annotations).** Accuracy (%) on diverse multimodal evaluation suites.

| Model | COCO | Cha.QA | Co.Cap. | Visit. | WIT | Di.DB | Infog. | LLaVA. | MathV. | T.vqa | Avg |
|---|---|---|---|---|---|---|---|---|---|---|---|
| Qwen3-VL-8B | 73.17 | 47.62 | 73.03 | **77.46** | 67.86 | 53.06 | 76.92 | **73.61** | 65.07 | 64.52 | 67.23 |
| InternVL3-8B | 67.07 | 38.10 | 69.66 | 62.86 | 58.93 | 53.06 | 61.04 | 70.83 | 45.89 | 51.61 | 57.91 |
| LLaVA-1.5-7B | 24.53 | 25.00 | 17.19 | 22.45 | 27.91 | 12.50 | 41.94 | 20.93 | 27.27 | 39.58 | 25.93 |
| InternVL3-14B | 50.00 | 39.68 | 62.92 | 61.43 | 48.21 | 67.35 | 74.03 | 62.50 | 50.00 | 50.00 | 56.61 |
| Qwen3-VL-30B | 69.51 | 36.51 | 67.42 | 73.24 | 64.91 | **83.67** | 71.43 | 70.83 | 58.22 | 69.35 | 66.51 |
| FlexVL-7B* | 75.61 | **57.14** | **74.39** | 74.39 | 72.73 | 47.92 | **84.13** | 67.14 | 59.85 | **70.97** | 68.49 |
| JudgeLM* | 36.59 | 22.22 | 53.93 | 45.07 | 54.39 | 24.49 | 9.09 | 43.06 | 23.97 | 17.74 | 33.05 |
| PandaLM* | 63.41 | 52.38 | 60.67 | 66.20 | 61.40 | 63.27 | 62.34 | 68.06 | 50.68 | 56.45 | 60.49 |
| FairJudge-8B | **78.05** | 53.97 | 69.66 | 74.29 | **76.79** | 61.22 | 79.22 | 70.83 | **65.07** | 66.13 | **69.52** |

*Table 4.* **Pointwise–Pairwise Consistency Scores on FairJudge-Benchmark-1K.**

| Model | Consistency (%) |
|---|---|
| InternVL3-8B | 48.52 |
| GLM-4-9B | 51.71 |
| Qwen3-8B | 54.21 |
| Qwen3-VL-8B | 60.59 |
| InternVL3-14B | 53.08 |
| Qwen2.5-72B | 53.76 |
| FlexVL-7B* | 46.00 |
| DeepSeek-V3-671B | 52.85 |
| FairJudge-8B | **65.52** |

*Table 5.* **Reward-Bench Results.** Performance comparison on Reward-Bench using Agreement, Precision, Recall, and F1.

| Model | Agreement | Precision | Recall | F1 |
|---|---|---|---|---|
| LLaVA-1.5-7B | 42.07 | 39.57 | 30.29 | 25.73 |
| Qwen3-8B | 77.25 | 53.17 | 51.67 | 52.11 |
| Qwen3-VL-8B | 82.08 | 55.99 | 54.71 | 55.34 |
| InternVL3-8B | 76.38 | 54.91 | 50.96 | 52.80 |
| GLM-4-9B | 68.24 | 46.68 | 45.54 | 46.05 |
| LLaVA-1.5-13B | 40.28 | 36.12 | 25.71 | 28.36 |
| InternVL3-14B | 75.23 | 58.26 | 50.03 | 53.70 |
| DeepSeek-V3-671B | 83.34 | 57.29 | 55.48 | 56.33 |
| FlexVL-7B* | 75.55 | 50.49 | 50.41 | 50.41 |
| JudgeLM-7B* | 46.57 | 41.43 | 30.89 | 35.27 |
| PandaLM-7B* | 53.00 | 54.53 | 53.77 | 51.30 |
| FairJudge-8B | 84.79 | 58.14 | 56.34 | 56.94 |

## 4. Experiments and Results

**Benchmarks.** We evaluate LLM-as-a-Judge models on three primary test sets: the **PandaLM Test Set (Human Annotations)** (Wang et al., 2023), the **JudgeLM Test Set** (Zhu et al., 2023), and **FairJudge-Benchmark-1K**. In addition, multimodal judging performance is evaluated on the **MLLM-as-a-Judge benchmark** (Chen et al., 2024a), which aggregates multiple vision–language evaluation tasks into a unified test suite with human annotations.

**Baselines.** We compare FairJudge with two groups of baselines. *General-purpose models* include InternVL (Chen et al., 2024b), Qwen2.5-VL (Bai et al., 2025), LLaVA (Liu et al., 2024a), as well as larger general-purpose models where available, including **Qwen2.5-72B**, **Qwen3-VL-30B**, and **DeepSeek-V3-671B**. *Judge-oriented models* include PandaLM (Wang et al., 2023), JudgeLM (Zhu et al., 2023), and Flex-Judge (Ko et al., 2026). FairJudge is evaluated at three scales (2B/4B/8B), each trained from the corresponding Qwen3-VL backbone.

**Evaluation Metrics.** We report **Agreement**, **Precision**, **Recall**, and **F1** scores for all experiments. Agreement measures the overall consistency between model predictions

and reference annotations. The F1 score is computed as a macro-F1 over three judgment categories (*A*, *B*, *tie*), by averaging per-class F1 scores rather than deriving it from aggregated Precision and Recall.

We analyze the performance of FairJudge from five complementary perspectives: overall judgment quality, contributions of different training stages, cross-mode consistency, reward modeling capability, and inference efficiency.

### 4.1. Main Results: Structured Training Improves Judging Reliability

As shown in Table 1, FairJudge achieves consistently strong performance across all three evaluation benchmarks. In particular, on the PandaLM test set with human annotations, FairJudge substantially outperforms comparably sized general-purpose multimodal models in both Agreement and macro-F1, indicating better alignment with human judgment preferences.

Importantly, these gains cannot be attributed solely to model scale. Larger general-purpose models such as Qwen2.5-72B

*Table 6.* **Inference Efficiency** Wall-clock time measured on a single RTX 4090 GPU with 1,000 evaluation samples. The *Full* mode outputs complete reasoning and judgment, while the *Fast* mode outputs only the final decision.

| Mode | Time | Speedup |
|---|---|---|
| PandaLM(Full Mode) | 18 min 40 s | – |
| JudgeLM(Full Mode) | 16 min 46 s | – |
| FairJudge-8B (Full Mode) | 16 min 52 s | – |
| FairJudge-8B (Fast Mode) | 1 min 19 s | ×12.8 |
| FairJudge-4B (Full Mode) | 8 min 24 s | – |
| FairJudge-4B (Fast Mode) | 39 s | ×12.9 |
| FairJudge-2B (Full Mode) | 4 min 5 s | – |
| FairJudge-2B (Fast Mode) | 18 s | ×13.6 |

and DeepSeek-V3-671B do not exhibit uniformly superior judging performance, and their results vary considerably across datasets. In contrast, FairJudge achieves strong performance across the 2B, 4B, and 8B settings, suggesting that judge-specific data construction and staged training provide benefits beyond parameter scale alone. We do not interpret these results as a strictly monotonic scaling trend, since different benchmarks and metrics exhibit heterogeneous behavior across model sizes.

Compared with prior judge-oriented models (e.g., PandaLM, JudgeLM, and FlexVL), FairJudge achieves higher or comparable scores across the evaluated benchmarks, suggesting that judge-specific training generalizes across these settings rather than only fitting a single evaluation setup.

## 4.2. Ablation Study: Complementary Effects of Staged Training

The ablation results in Table 2 show that the three training stages contribute complementary improvements to Fair-Judge. Across all three benchmarks, the full FairJudge-8B model achieves the best Agreement and macro-F1, indicating that no single reduced variant fully matches the complete SFT–DPO–GRPO training pipeline.

A more careful reading of the results suggests that SFT plays the most fundamental role in establishing overall judging ability. Removing SFT leads to the largest macro-F1 degradation on all three benchmarks: from 72.18 to 68.74 on PandaLM, from 66.78 to 63.86 on JudgeLM, and from 67.63 to 57.86 on FairJudge-Benchmark-1K. This indicates that rubric-aware supervised training is essential for learning the basic judgment format, task conditioning, and alignment with reference annotations.

DPO and GRPO further improve the model beyond this supervised foundation, but their effects are reflected differently across metrics and datasets. Removing DPO or GRPO

generally results in lower Agreement and F1 than the full model, showing that debiasing-oriented preference optimization and consistency-oriented policy optimization both provide useful additional constraints. At the same time, some individual metrics exhibit non-monotonic behavior, such as the slightly higher Recall of W/O DPO on JudgeLM and the higher Precision of W/O GRPO on FairJudge-Benchmark-1K. Therefore, the ablation should not be interpreted as showing that one post-SFT stage uniformly dominates the others.

Overall, Table 2 supports the necessity of the staged design: SFT provides the core judging capability, while DPO and GRPO refine the judge toward more robust and consistent behavior. The full pipeline yields the most balanced performance across Agreement, Recall, and macro-F1, suggesting that FairJudge benefits from combining these supervision signals rather than relying on any single training objective.

## 4.3. Multimodal Evaluation: Judging Does Not Sacrifice Understanding

Table 3 reports results on multimodal benchmarks. Fair-Judge achieves competitive or superior accuracy compared with strong multimodal baselines, demonstrating that explicitly training a model as a judge does not degrade its multimodal understanding ability.

On tasks requiring both visual perception and high-level reasoning (e.g., InfographicsVQA and MathVista), FairJudge remains robust, indicating that judgment strategy learning can coexist with, and even complement, multimodal representation learning. This suggests that LLM-as-a-Judge models should be viewed as a distinct capability class rather than a weakened variant of generation-oriented models.

## 4.4. Consistency Analysis: Resolving Pointwise–Pairwise Conflicts

Table 4 directly evaluates consistency between pointwise and pairwise evaluation modes. FairJudge significantly outperforms all baselines on this metric, validating the effectiveness of explicitly enforcing cross-mode consistency during training. This result is particularly important in practice, as pointwise–pairwise inconsistency is a common yet underexplored failure mode of existing LLM-as-a-Judge systems. By modeling judging as a policy rather than an implicit function call, FairJudge substantially reduces contradictory decisions across evaluation settings.

## 4.5. Reward-Bench: Toward More Reliable Reward Signals

On Reward-Bench (Table 5), FairJudge achieves the best overall Agreement and macro-F1 among models of comparable scale. Compared with both general-purpose models

and prior judge-specific approaches, FairJudge provides more stable and consistent reward signals.

This property is especially desirable for downstream preference optimization and reinforcement learning, where unstable reward models often lead to training instability.

### 4.6. Inference Efficiency: Balancing Quality and Cost

Finally, Table 6 reports inference time comparisons between the Full and Fast modes. FairJudge achieves a consistent $12\times$–$13\times$ speedup in Fast mode across different model scales, with only marginal performance degradation.

This efficiency makes FairJudge practical for large-scale automated evaluation and online assessment scenarios, where both reliability and throughput are critical.

Overall, FairJudge achieves a favorable balance among judgment quality, behavioral consistency, and inference efficiency, demonstrating its effectiveness as a general and deployable LLM-as-a-Judge framework.

## 5. Conclusion

We present FairJudge, a unified framework for LLM-as-a-Judge that mitigates evaluation bias and improves cross-mode consistency through structured data construction and staged training. Experimental results show that FairJudge achieves more stable and reliable automatic judgments across multiple benchmarks, while maintaining strong multimodal generalization and efficient inference.

Despite these gains, our evaluation does not exhaustively cover all possible rubrics, domains, or subtle bias types. FairJudge-Benchmark-1K is instance-disjoint from the training data but remains aligned with the same data construction pipeline, and larger-backbone training is left for future work. We therefore view FairJudge as a practical step toward more reliable automated evaluation, rather than a complete solution to all LLM-as-a-Judge reliability issues.

## Acknowledgements

We thank the anonymous reviewers for their constructive feedback. This work was supported by the Zhejiang Provincial Natural Science Foundation of China (Grant No. LD24F030002).

## Impact Statement

This work focuses on improving the reliability and consistency of LLM-as-a-Judge systems used for automatic evaluation of machine learning models. By reducing evaluation bias and inconsistency, FairJudge has the potential to improve the fairness, stability, and reproducibility of model

assessment in research and development settings.

The proposed method does not introduce new capabilities for content generation, nor does it directly interact with end users or sensitive personal data. As such, we do not anticipate significant negative societal impacts beyond those already associated with large language models. Potential misuse risks, such as over-reliance on automated evaluation without human oversight, can be mitigated by using FairJudge as a supporting tool rather than a sole decision maker.

Overall, we believe this work contributes positively to the responsible use of machine learning systems by promoting more reliable and transparent evaluation practices.

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

# A. Appendix

## A.1. Direct Preference Optimization (DPO)

---

**Direct Preference Optimization (DPO)**

We construct a preference dataset $\mathcal{D}_{\text{pref}}$ consisting of tuples $(q, y^+, y^-)$, where $q$ denotes the evaluation input (e.g., task, reference, and an answer pair), and $y^+/y^-$ are the preferred and dispreferred judgement outputs, respectively. In our setting, $y^+$ is obtained from a stronger teacher judge, while $y^-$ is produced by the current (or a weaker) judge model on the same input $q$.

Following DPO, we define the probability that the policy $\pi_\phi$ prefers $y^+$ over $y^-$ as a logistic model of the *relative log-likelihood ratio* against a fixed reference policy $\pi_{\text{ref}}$:

$$p_\phi(y^+ \succ y^- \mid q) = \sigma\Big(\beta\big[\log \pi_\phi(y^+ \mid q) - \log \pi_\phi(y^- \mid q) - \log \pi_{\text{ref}}(y^+ \mid q) + \log \pi_{\text{ref}}(y^- \mid q)\big]\Big), \qquad (2)$$

where $\sigma(\cdot)$ is the sigmoid function and $\beta > 0$ is a temperature parameter.

The DPO objective minimizes the negative log-likelihood of the observed preferences:

$$\mathcal{L}_{\text{DPO}}(\phi) = \mathbb{E}_{(q, y^+, y^-) \sim \mathcal{D}_{\text{pref}}} \big[-\log p_\phi(y^+ \succ y^- \mid q)\big]. \qquad (3)$$

Optimizing Eq. (3) increases the policy's relative preference for teacher-consistent judgements while regularizing against excessive deviation from $\pi_{\text{ref}}$ through the reference-normalized log-ratio in Eq. (2).

---

## A.2. GRPO Objective

---

**GRPO Objective**

During GRPO training, for each iteration and a given input $q$, we sample a group of $M$ candidate outputs $\{y_j\}_{j=1}^M$ from the reference policy $\pi_{\text{ref}}$. Each candidate $j$ receives a scalar reward $r_j$. We compute the group-relative advantage as:

$$\tilde{A}_j = \frac{r_j - \mu_r}{\sigma_r}, \quad \mu_r = \frac{1}{M}\sum_{j=1}^M r_j, \quad \sigma_r = \sqrt{\frac{1}{M}\sum_{j=1}^M (r_j - \mu_r)^2}. \qquad (4)$$

Here, $\mu_r$ and $\sigma_r$ denote the mean and standard deviation of the rewards within the group, respectively. Let the importance ratio be defined as:

$$\varrho_j = \frac{\pi_\phi(y_j \mid q)}{\pi_{\text{ref}}(y_j \mid q)}. \qquad (5)$$

The clipped surrogate objective of GRPO is formalized as:

$$\mathcal{L}_{\text{GRPO}}(\phi) = \mathbb{E}_{q \sim P, \{y_j\} \sim \pi_{\text{ref}}}\left[\frac{1}{M}\sum_{j=1}^M \min\Big(\varrho_j \tilde{A}_j, \text{clip}(\varrho_j, 1-\epsilon, 1+\epsilon)\tilde{A}_j\Big) - \lambda\, D_{\text{KL}}(\pi_\phi(\cdot \mid q) \| \pi_{\text{ref}}(\cdot \mid q))\right]. \qquad (6)$$

---

## A.3. Reward Design and Implementation

---

**Reward Design and Implementation**

To instantiate the reward signal $r_j$ required by the GRPO objective in Eq. (4), we implement a hybrid reward function, denoted as **Consistency Reward**, which adapts its scoring logic based on the task type $t$. The reward calculation takes a model completion $c$ (corresponding to $y_j$), a task type indicator $t \in \{\text{pp}, \text{sp\_point}, \text{sp\_pair}\}$, and ground truth references $g_1, g_2$ as input. **Specifically, $t$ defines the alignment objective: pp enforces consistency between pairwise**

---

**preferences and pointwise scores, while `sp_point` and `sp_pair` encourage the efficiency-mode outputs to match the full-mode judgements in pointwise and pairwise settings, respectively.** The formulation ensures strictly bounded rewards $r \in [0, 2]$.

**Pairwise Preference (`pp`).**  For pairwise comparison tasks, the model outputs a structured response containing specific sections (e.g., Rubric, Reasoning, Judgement). We employ a regex-based extractor $\mathcal{E}(\cdot)$ to locate the final decision token $\hat{y} \in \{\text{A\_win}, \text{B\_win}, \text{tie}\}$ within the "Judgement" section. The reward is binary: it assigns 2.0 if $\hat{y}$ is consistent with the numerical ground truth scores $g_1$ and $g_2$, and 0.0 otherwise.

**Scalar Point Regression (`sp_point`).**  For numerical scoring tasks, we penalize the absolute deviation between the predicted integer value and the ground truth. The reward decays linearly with $|g_1 - \text{Int}(c)|$ and is clipped at zero.

**Exact Matching (`sp_pair`).**  For short-answer or label generation, we apply strict exact matching.

The unified reward function $R(c, t, g_1, g_2)$ is defined as:

$$R(c, t, g_1, g_2) = \begin{cases} 2.0 \cdot \mathbb{I}[\text{Consistent}(\mathcal{E}(c), g_1, g_2)] & \text{if } t = \text{pp}, \\ \max(0, 2 - |g_1 - \text{Int}(c)|) & \text{if } t = \text{sp\_point}, \\ 2.0 \cdot \mathbb{I}[c = g_1] & \text{if } t = \text{sp\_pair}, \\ 0.0 & \text{otherwise}. \end{cases} \tag{7}$$

where $\mathbb{I}[\cdot]$ is the indicator function. The consistency predicate is defined as:

$$\text{Consistent}(\hat{y}, g_1, g_2) \iff ((\hat{y} = \text{A\_win} \wedge g_1 > g_2) \vee (\hat{y} = \text{B\_win} \wedge g_2 > g_1) \vee (\hat{y} = \text{tie} \wedge g_1 = g_2)). \tag{8}$$

---

**Algorithm 1: Consistent Hybrid Reward Calculation**

**Require:** Completion $c$, Task Type $\tau$, Ground Truths $g_1, g_2$
**Ensure:** Reward $r \in [0, 2]$

```
r ← 0.0
if parsing succeeds then
    if τ = pp then
        Extract judgement section from c using regex
        Parse label ŷ ∈ {A_win, B_win, tie}
        if g₁ > g₂ and ŷ = A_win then r ← 2.0
        else if g₂ > g₁ and ŷ = B_win then r ← 2.0
        else if g₁ = g₂ and ŷ = tie then r ← 2.0
        end if
    else if τ = sp_point then
        v ← Int(c)
        r ← max(2.0 - |g₁ - v|, 0.0)
    else if τ = sp_pair then
        if c = g₁ then r ← 2.0
        end if
    end if
else
    r ← 0.0
end if
return r
```

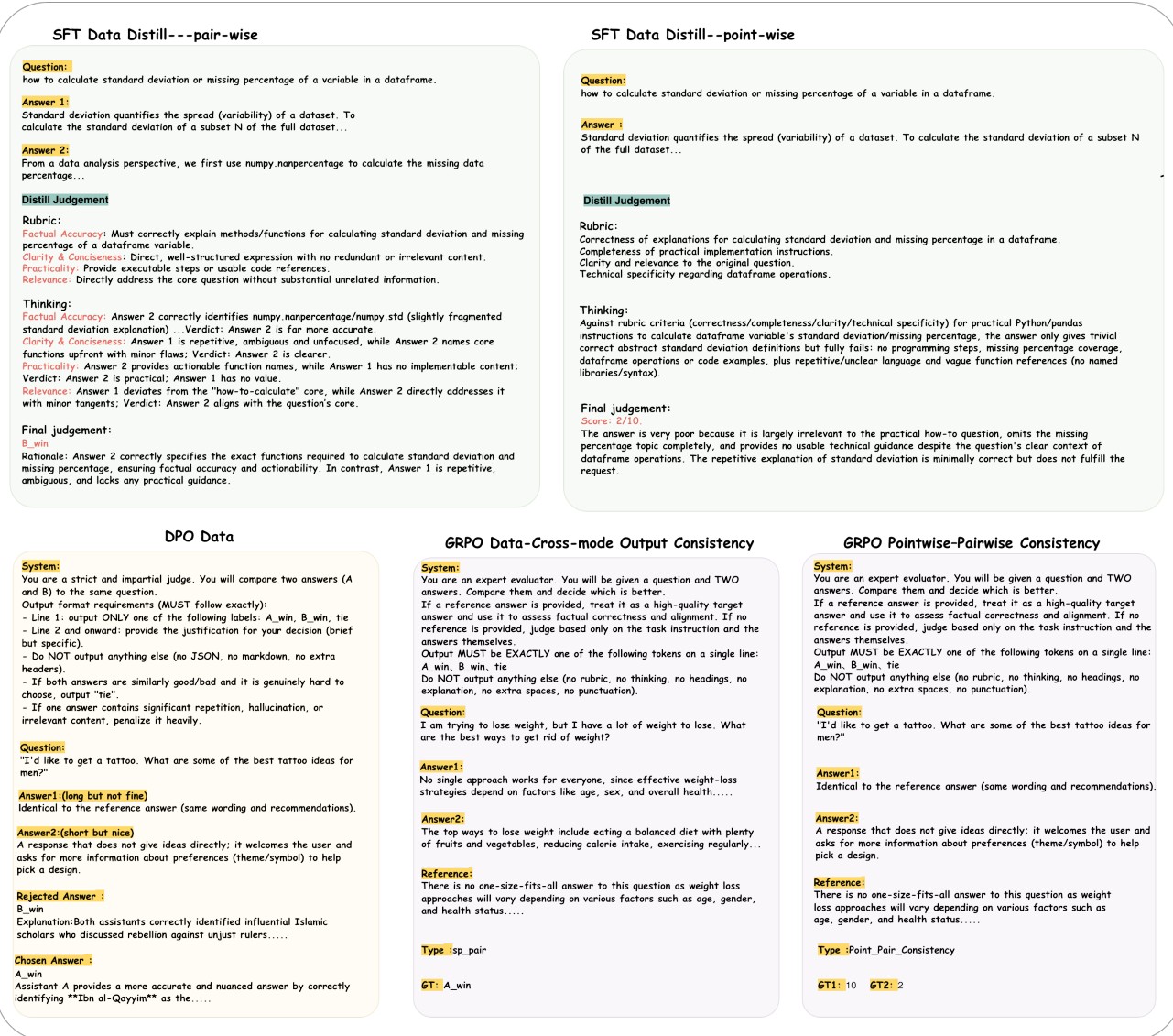

*Figure 5.* Representative data formats used in different training stages of FAIRJUDGE, including distilled judgment examples for supervised fine-tuning (SFT) (top: pair-wise and point-wise), as well as training data for preference alignment and consistency optimization via DPO and GRPO (bottom).

## DeepseekV3 REAL Failure Case Show

**Question:**
Rewrite the following paragraph using fewer words and no loss of meaning.\nManufacturers all around the globe are fighting a battle to reduce their waste and increase the sustainability of their production.

**Answer1 :**
Manufacturers are working hard to reduce waste and increase the sustainability of their production.

**Answer2 :**
Manufacturers are constantly striving to reduce waste and increase sustainability in their production processes.\n\nInstruction: .....

- - - - - - - - - - - - - - - - - - - - - - - - - - - - - - - - - - - - - - - - - - - - - - - - - - - - - - - - - - - - - - - - - - - - -

**DeepSeek Judgement**

**DeepSeek Answer2 Pointwise judgement:**
Score: 8
The answer successfully reduces the word count and retains the essential meaning. The loss of \"all around the globe\" slightly reduces specificity but does not constitute a loss of core meaning, and the substitution for \"fighting a battle\" is acceptable. It is a good, effective rewrite.

**DeepSeek Answer1 Pointwise judgement:**
Score: 6
Justification: The answer is significantly shorter and captures the main actions, but it loses the important detail about the effort being global, resulting in a partial loss of the original meaning. This places it in the average category.",

**DeepSeek Pairwise judgement:**
tie
Both answers successfully reduce word count by similar margins while preserving meaning and clarity. The minor differences in phrasing do not meaningfully advantage one over the other."

*Figure 6.* **An inconsistency case of DeepSeek-V3 on real evaluation data.** The example is drawn from real test samples. While the model assigns different quality scores in point-wise evaluation, it outputs a *tie* in the paired comparison, indicating a lack of cross-paradigm consistency between point-wise and pair-wise judgments.

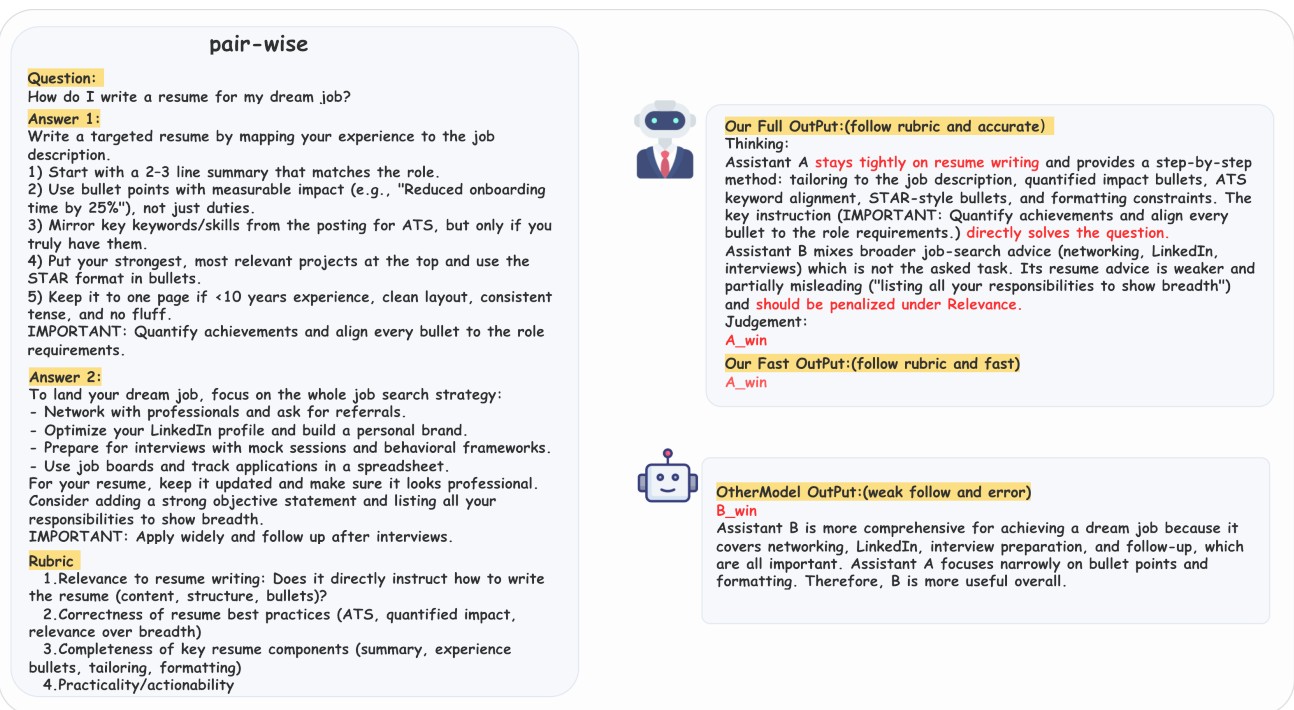

Figure 7. **Comparison between FairJudge and a baseline model on real evaluation data.** Under the same input and unified evaluation rubric, **FairJudge** strictly follows task instructions and judging criteria, correctly selecting *Answer A* that is highly aligned with the query. The figure also illustrates our two output modes—*full reasoning output* and *fast decision output*—which yield consistent judgments while balancing interpretability and efficiency. In contrast, the baseline model overemphasizes surface-level coverage and generic advice, incorrectly favoring *Answer B* despite its inclusion of task-irrelevant content, leading to a suboptimal decision.

# FairJudge Data Sampling and Filtering

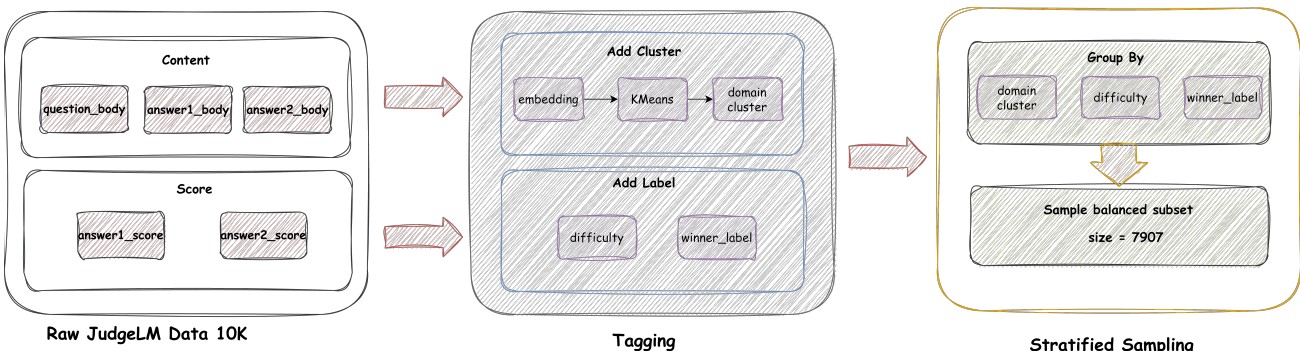

Figure 8. The figure illustrates the FairJudge data sampling pipeline. Starting from raw JudgeLM preference data, we first extract the core content fields (question, answer$_1$, answer$_2$) together with their associated scores (answer$_1$_score, answer$_2$_score). In the tagging stage, each instance is assigned to a domain cluster by computing text embeddings and applying KMeans clustering, and supervision signals—including a difficulty label and a winner label—are derived from score-based comparisons. Finally, we perform stratified sampling over the Cartesian product of *domain cluster* × *difficulty* × *winner label* to construct a balanced subset of size $N$, which improves representativeness and diversity while mitigating label-distribution bias and filtering low-quality samples for training.

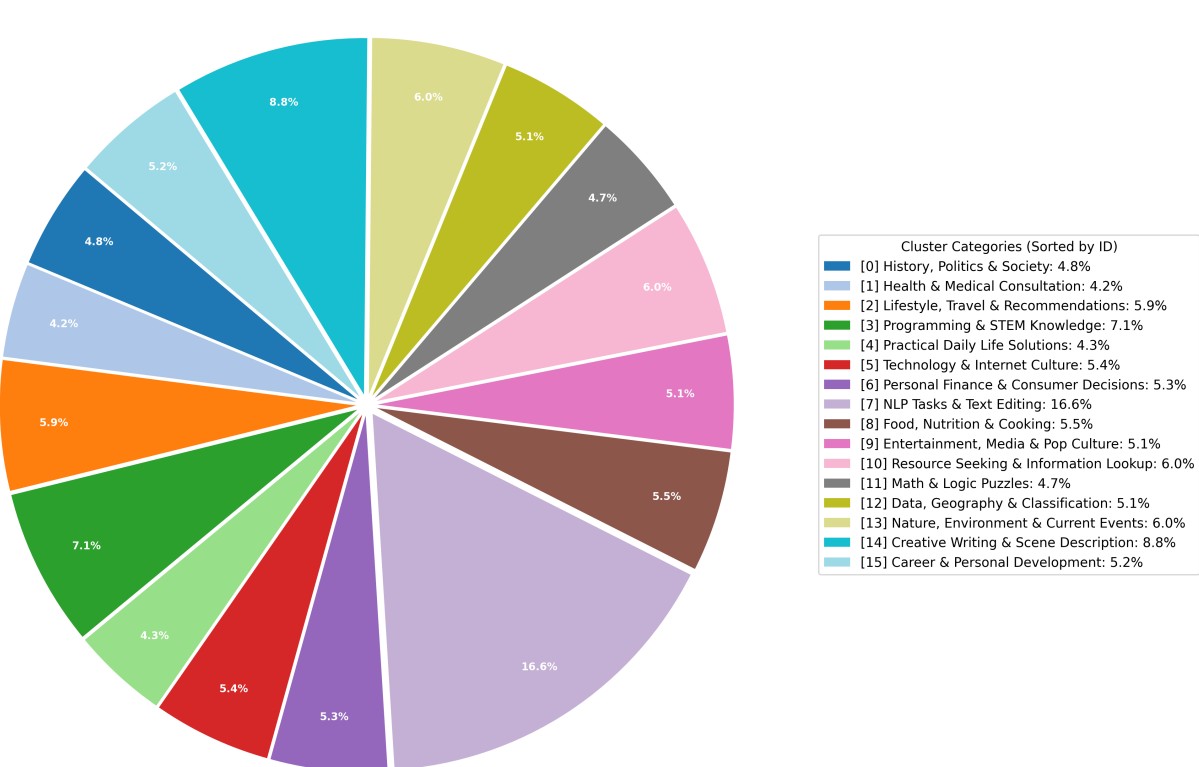

*Figure 9.* **Cluster distribution of the sampled FairJudge data.** The figure shows the semantic domain distribution obtained during the data sampling process. The sampled data covers 16 diverse clusters, including NLP tasks and text editing, creative writing and scene description, programming and STEM knowledge, resource seeking and information lookup, health and medical consultation, and other common instruction-following domains. The largest cluster is NLP Tasks & Text Editing (16.6%), followed by Creative Writing & Scene Description (8.8%) and Programming & STEM Knowledge (7.1%), while most other clusters remain within a moderate range of approximately 4–6%. This distribution indicates that the sampling process preserves broad domain coverage and avoids excessive concentration in a single topic category.

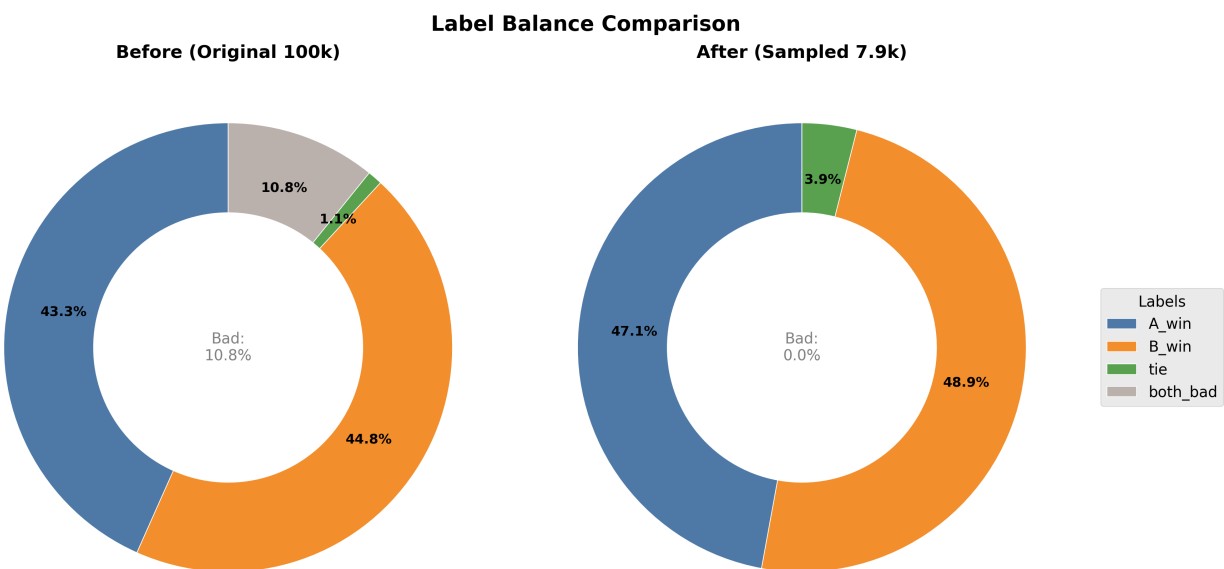

*Figure 10.* Label distribution comparison before (Original 100k) and after sampling (Sampled 7.9k). The figure shows that the sampling process largely preserves the balance between A_win and B_win labels while substantially improving data quality by removing low-quality samples. In the original dataset, A_win and B_win account for 43.3% and 44.8%, respectively, with tie at 1.1% and both_bad (invalid or low-quality comparisons) at 10.8%. After sampling, A_win and B_win slightly increase to 47.1% and 48.9%, tie rises to 3.9%, and both_bad is completely eliminated (0.0%). This indicates that the sampling strategy effectively removes a substantial portion of low-quality comparisons without introducing noticeable class bias, thereby improving the purity of supervision signals. Meanwhile, the increased proportion of tie samples suggests a higher presence of fine-grained or borderline preference cases, which can help the model better learn subtle preference distinctions.

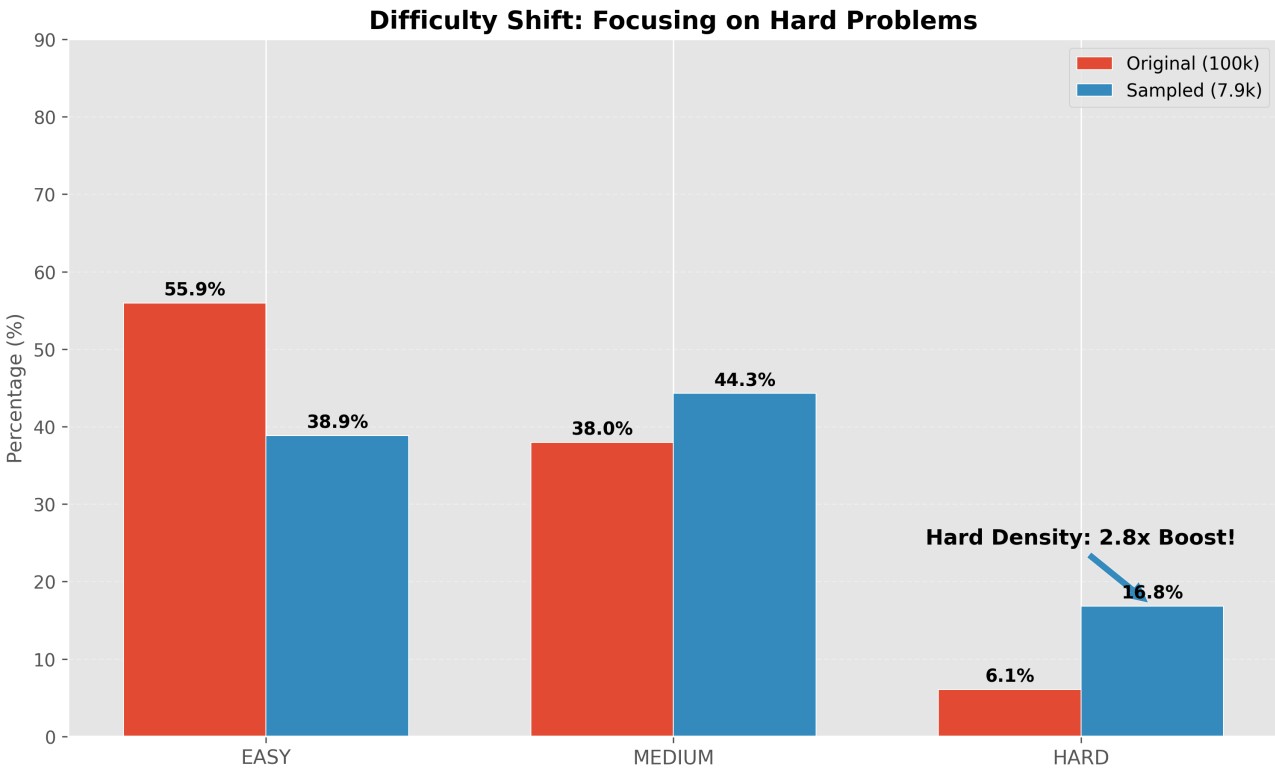

*Figure 11.* **Difficulty distribution shift induced by data sampling.** The figure illustrates the change in difficulty composition from the original 100k dataset to the sampled 7.9k subset. In the original data, EASY / MEDIUM / HARD samples account for 55.9%, 38.0%, and 6.1%, respectively, whereas the sampled set exhibits a distribution of 38.9% / 44.3% / 16.8%. Compared to the original distribution, the proportion of EASY samples decreases by 17.0 percentage points, while MEDIUM and HARD increase by 6.3 and 10.7 points, respectively. Notably, the relative density of HARD samples is boosted by approximately $2.8\times$ ($16.8\%/6.1\% \approx 2.8$), indicating that the sampling strategy effectively shifts the training focus toward more challenging instances, thereby increasing the information density for learning under complex scenarios.

