# OpenReview forum: "FairJudge : An Adaptive, Debiased, and Consistent LLM-as-a-Judge"
_ICML.cc/2026/Conference — ICML 2026 regular_

### Official Review · Reviewer_otcA · 2026-03-12

**Soundness:** 3
**Presentation:** 3
**Significance:** 3
**Originality:** 2
**Overall Recommendation:** 4
**Confidence:** 3

**Summary:**

This paper proposes FairJudge, a framework that treats LLM judging behavior as a learnable policy rather than a fixed function. It targets three properties, namely adaptivity to task-specific rubrics, robustness to non-semantic biases such as position and length, and consistency across pointwise and pairwise evaluation modes. A high-information-density dataset is constructed with explicit rubric conditioning, contrastive debiasing pairs, and cross-mode consistency supervision. Training follows a three-stage curriculum in which SFT establishes rubric-aware judging, DPO reduces bias, and GRPO enforces cross-mode consistency. Experiments on multiple benchmarks show improvements in agreement and F1, reduced bias, and strong cross-mode consistency, with the 8B model outperforming much larger general-purpose LLMs on judging tasks.

**Compliance With Llm Reviewing Policy:**

Affirmed.

**Final Justification:**

The rebuttal addressed my concerns. I maintain my original score.

**Key Questions For Authors:**

1. Can you evaluate FairJudge on a task with a completely novel rubric not represented in the training data? For example, judging code security or academic paper novelty. This would directly test the adaptivity claim.
2. After GRPO training, does the proportion of "tie" judgments increase? If the model learns to default to tie to maximize consistency scores, this would represent a degenerate solution rather than genuine improvement.
3. What happens if you apply the FairJudge training recipe to a larger backbone? Would this invalidate the "structure over scale" narrative, or do the gains still hold?
4. The paper claims that removing GRPO causes the most significant drop, but the F1 numbers in Table 2 show that removing SFT consistently causes larger drops across all three benchmarks. Can you clarify which training stage is truly most critical and reconcile this discrepancy?

**Limitations:**

Yes

**Strengths And Weaknesses:**

Strengths
1. The policy-based formulation is a meaningful conceptual advance. Explicitly modeling judging as a conditional decision policy rather than a fixed scoring function provides a principled foundation for addressing adaptivity, debiasing, and consistency jointly. This reframing clarifies what previous approaches implicitly assumed but did not constrain.
2. The three-stage training design is well-motivated and complete. Each stage targets a distinct behavioral property, and the ablation study confirms that each contributes. The curriculum ordering is sensible, establishing base capability first and then adding constraints progressively.
3. Cross-mode consistency is identified as an important and underexplored training objective. Most prior work on judge training focuses on accuracy or bias reduction. Explicitly optimizing for pointwise-pairwise agreement via GRPO highlights a meaningful direction, regardless of whether the ablation fully supports the paper's claim about its relative importance (see weakness 6).
4. The results are evaluated across diverse benchmarks with multiple baselines. The paper includes both judge-specific and general-purpose baselines at various scales, human-annotated test sets, multimodal benchmarks, and Reward-Bench. This breadth of evaluation is a strength.

Weaknesses
1. Adaptivity, the first of the three claimed properties, is the least validated. The training data is built from a single source. All evaluation benchmarks share similar task distributions. There is no test of whether FairJudge can follow a truly novel rubric at test time, such as one involving a new domain or evaluation criterion not seen during training. Without such a test, the adaptivity claim is not convincingly supported.
2. The debiasing scope may be limited to easy cases. The paper addresses position, length, format, and style biases. These are important but relatively straightforward non-semantic factors. Subtler biases such as authority bias, fluency bias, or self-preference bias are not discussed. It is worth clarifying whether the method addresses only surface-level biases or deeper judgment distortions.
3. The consistency metric deserves more scrutiny. The best consistency score is around 65%, meaning roughly one in three evaluations still produces contradictory pointwise and pairwise judgments. Some degree of inconsistency between modes may actually be reasonable, since the two modes can capture different aspects of quality. The paper does not discuss whether 100% consistency is desirable or whether some inconsistency is legitimate.
4. The comparison with large models needs qualification. The paper argues that structure matters more than scale based on the observation that FairJudge-8B outperforms certain 72B and 671B models on judging tasks. However, those larger models are general-purpose and not fine-tuned for judging. The comparison shows that task-specific training helps, which is unsurprising. A fairer test of the scale claim would be to apply FairJudge's training method to a larger backbone and see if gains persist.
5. Training data leakage risk from shared data sources. FairJudge-16K and the JudgeLM test set both derive from the same data source. While instance-level separation is claimed, distributional similarity is inevitable. Performance on the independently sourced PandaLM test set may be a more reliable indicator of true generalization.
6. The ablation claim that GRPO contributes the most is not well supported by the data. The paper states that removing GRPO results in the most significant drop. However, the F1 numbers in Table 2 show that removing SFT causes larger drops on all three benchmarks, and the same pattern holds for the majority of other metrics. This discrepancy between the stated conclusion and the ablation data undermines confidence in the analysis and raises the question of which training stage is truly most critical.

---

> ### Author Rebuttal · Authors · 2026-03-31
>
> **We'd like to express our sincere gratitude for your careful readings and valuable comments.** We provide our feedback as follows:
>
> ### **Response to Weakness1 & Question 1**
>
> **Clarification on new rubric dataset test.**
> **Thanks for the valuable comment**. Due to the character limit, we have included the relevant additional experiments in other responses: please refer to our reply to **reviewer meQk (W1)** for the adaptivity-related evaluation.
>
>
> ---
> ---
>
> ### **Response to W2**
>
> **Clarification on new debias dataset test.**
> **Thanks for the valuable comment**. Due to the character limit, we have included the relevant additional experiments in other responses: please refer to our reply to **reviewer 65Ut (W2)** for the debiasing-related evaluation.
>
> ---
> ---
>
> ### **Response to W3 & Q2**
>
> **Thanks for the valuable comment**. Our goal is not to achieve superficial 100% mode agreement, nor to enforce pointwise and pairwise judgments to be strictly identical in all cases. More precisely, we aim to reduce **direct logical conflicts** under aligned rubrics and the same evaluation content. Therefore, we will clarify in the revision that residual inconsistencies do not necessarily indicate errors, and certain mode-specific differences can be reasonable.
>
> On the other hand, defaulting to *tie* is not a viable strategy under the GRPO reward. In the pairwise setting, a *tie* is rewarded only when \( g_1 = g_2 \); predicting *tie* for non-tie samples yields zero reward. Thus, the reward design does not encourage a degenerate “trading ties for consistency” strategy.
>
>
> To address the concern about potential tie inflation, we explicitly measure the proportion of tie predictions before and after GRPO:
>
> | Model | JudgeBench | JudgeLM | PandaLM | SelfEval |
> |---|---:|---:|---:|---:|
> | Before GRPO | 0.0161 | 0.0802 | 0.1429 | 0.0456 |
> | After GRPO  | 0.0145 | 0.0770 | 0.1228 | 0.0481 |
>
> We observe that the tie rate decreases on three out of four benchmarks, with only a negligible increase on SelfEval. This suggests that GRPO does not lead to a degenerate strategy of predicting more ties. Instead, the consistency improvements are achieved without relying on tie inflation, indicating more reliable and meaningful alignment across evaluation modes.
>
>
>
> ---
> ---
>
> ### **Response to W4 & Q3**
> **Thanks for the valuable comment**. We agree that a more direct test would be to apply our training method to larger backbones. Due to practical constraints, we have not conducted such experiments in the current work, and leave this as future work.
>
> That said, we observe consistent improvements across multiple backbone sizes (2B, 4B, and 8B), suggesting that the effectiveness of our approach is not limited to a specific scale. We therefore position our claim more cautiously as: judge-specific training yields gains that are not solely explained by parameter scale.
>
>
> ---
> ---
>
> ### **Response to W5**
>
>
>
> **Thank you for raising this important comment**. **PandaLM is independently collected and serves as a human-annotated benchmark**. FairJudge demonstrates consistent and stable improvements on PandaLM, following similar trends as observed on the other test sets. This suggests that the performance gains are not tied to a specific data source or construction pipeline.
>
> In addition, we performed an explicit **hash overlap check** between our training set and the JudgeLM test set, and observed **zero overlap**, confirming that there are no instance-level duplicates.
>
> Taken together, the consistent improvements on an independent benchmark (PandaLM), along with the absence of instance-level overlap, indicate that the gains of FairJudge are not driven by data leakage, but generalize across different data sources.
>
>
> ---
> ---
>
> ### **Response to W6 & Q4**
> **Thanks for the valuable comment**.  We agree that, when considering overall F1 in Table 2, SFT plays a more fundamental role in overall performance. The wording in the original manuscript may not have been sufficiently precise. A more accurate interpretation is that SFT establishes the basic judging capability and rubric adherence, while DPO and GRPO provide additional improvements on their respective target properties, with GRPO contributing more directly to cross-mode consistency rather than being reflected solely through overall F1. We will clarify this point in the revision.
>
> **We sincerely thank the reviewer for your careful reading and thoughtful feedback.** We hope our clarification contributes to a clearer and more complete understanding of our work.

---

> > ### Author Rebuttal · Reviewer_otcA · 2026-04-04
> >
> > The authors have adequately addressed the specific concerns raised in my review.

---

> > > ### Author Response · Authors · 2026-04-04
> > >
> > > Thank you very much for your time and for carefully reviewing our rebuttal.
> > > We sincerely appreciate your acknowledgment that the concerns have been adequately addressed.
> > >
> > > We are glad that the additional clarifications and experiments helped resolve the issues raised in your original review. Your feedback has been invaluable in improving the quality and clarity of our work.
> > >
> > > Thank you again for your support.

---

### Official Review · Reviewer_65Ut · 2026-03-12

**Soundness:** 2
**Presentation:** 2
**Significance:** 2
**Originality:** 2
**Overall Recommendation:** 4
**Confidence:** 3

**Summary:**

This paper proposes FairJudge, a learnable LLM-as-a-Judge framework designed to improve automatic evaluation. It argues that existing judge models suffer from limited adaptivity to task-specific criteria, systematic non-semantic biases, and inconsistency across evaluation modes. To address this, the paper models judging as a conditional decision policy rather than a fixed scoring function. It further constructs a high-information-density training dataset and trains the judge with a curriculum-style SFT–DPO–GRPO pipeline to improve rubric adherence, debiasing, and cross-mode consistency.

**Compliance With Llm Reviewing Policy:**

Affirmed.

**Final Justification:**

Score increased from 3 to 4

**Key Questions For Authors:**

Please see Weaknesses.

**Limitations:**

Please see Weaknesses.

**Strengths And Weaknesses:**

**Summary of Strengths:**

* The paper is well presented and easy to follow. The content is rich, and the figures are clear and informative, which makes the overall method and motivation easy to understand.

* The paper is well organized. The structure is clear, with a logical progression from problem formulation to data construction, training design, and experimental validation.

* The method is conceptually well motivated. The paper provides a coherent perspective by formulating judging as a learnable policy and aligning the training pipeline with the three target properties: adaptivity, debiasing, and consistency.

**Summary of Weaknesses:**

* The experimental evidence is not fully consistent, and some comparisons are not sufficiently controlled. In Figure 1, FairJudge-2B and FairJudge-4B each perform better on different metrics, while one would generally expect the 4B model to consistently dominate the 2B model. Moreover, FairJudge-8B is compared against FlexVL, JudgeLM, and PandaLM 7B models, which introduces a parameter-scale mismatch and makes the comparison less conclusive. The performance trends across datasets are also somewhat difficult to interpret. More size-matched and carefully controlled comparisons would strengthen the empirical claims.

* The debiasing gains may be partly tied to the specific perturbation patterns used during data construction. Since the debiasing stage relies on structured perturbations, the model may primarily learn invariance to the particular non-semantic factors seen during training, rather than acquiring a more general debiasing capability. It remains unclear whether the method would generalize to unseen or naturally occurring bias types.

* The GRPO stage may be vulnerable to reward hacking. As with many RL-based objectives, if the consistency reward is not carefully specified, the model may exploit superficial strategies that improve the reward without genuinely improving judgment quality. The paper would be stronger if it included failure cases or qualitative analyses showing whether such behavior occurs in practice.

* The paper lacks theoretical analysis. While the method is intuitively motivated, the paper does not provide a deeper theoretical justification for why the staged SFT–DPO–GRPO pipeline should reliably improve adaptivity, debiasing, and cross-mode consistency, nor does it analyze the potential interactions or conflicts among these objectives.

---

> ### Author Rebuttal · Authors · 2026-03-31
>
> **We'd like to express our sincere gratitude for your careful readings and valuable comments.** We provide our feedback as follows:
>
> ### **Response to Weakness1**
>
> **Clarification on the 2B–4B scaling claim.**
> **Thanks for the valuable comment**. Our goal is to show that the FairJudge training recipe transfers across scales and enables a quality–efficiency trade-off, not to claim strictly monotonic scaling across all benchmarks and metrics. Given benchmark heterogeneity and the greater non-monotonicity of smaller models under fixed budgets, we will temper this claim and avoid over-interpreting the 2B/4B results as direct evidence for “structure over scale.”
>
> **Size-matched validation at the 7B scale.**
> We further conduct strictly size-matched experiments by training FairJudge on **7B backbones** with the same architecture family as our original base models (Qwen series). FairJudge-7B achieves stronger Agreement and competitive overall performance compared to existing 7B judge baselines. This indicates that the performance gains are **not merely due to the larger 8B parameter scale**, but remain valid **under the same-scale backbone setting**.
>
> **Metric notation.**
> JL / PL / SE denote evaluation on JudgeLM / PandaLM / SelfEval, respectively.
> A / P / R denote Agreement / Precision / Recall.
>
> | Model | JL-A | JL-P | JL-R | JL-F1 |  | PL-A | PL-P | PL-R | PL-F1 |  | SE-A | SE-P | SE-R | SE-F1 |
> |:--|--:|--:|--:|--:|:--:|--:|--:|--:|--:|:--:|--:|--:|--:|--:|
> | PandaLM-7B | 67.42 | 44.99 | 48.53 | 46.66 |  | 65.82 | 44.56 | 48.93 | 46.24 |  | 52.46 | 35.00 | 39.23 | 36.99 |
> | FlexVL-7B | 77.44 | 64.82 | 57.49 | 57.42 |  | 71.34 | 64.30 | 62.18 | 63.00 |  | 59.58 | 49.56 | 46.01 | 44.72 |
> | JudgeLM-7B | 78.00 | 66.02 | **67.61** | 66.66 |  | 66.77 | 63.83 | **71.95** | 63.92 |  | 69.56 | 65.27 | 67.41 | 66.11 |
> | FairJudge(Qwen2.5-7B-VL) | 79.03 | **76.41** | 59.48 | 59.96 |  | 75.03 | **69.49** | 65.82 | 67.01 |  | 69.72 | **73.61** | 59.90 | 62.82 |
> | FairJudge(Qwen2.5-7B-Omni) | **79.51** | 71.41 | 60.37 | **61.04** |  | **75.77** | 68.56 | 68.72 | **68.53** |  | **70.42** | 68.68 | **70.09** | **69.10** |
>
> ---
> ---
>
> ### **Response to W2**
>
> | Model | JudgeBench-Claude Accuracy | JudgeBench-GPT-4o Accuracy |
> |:--|--:|--:|
> | Qwen3-VL-8B | 0.5593 | 0.5943 |
> | FairJudge-8B | **0.6333** | **0.6600** |
>
> **We thank the reviewer for raising this important point.** We would like to clarify that our debiasing evaluation is not limited to only a few manually constructed bias types in the paper. Beyond the controlled experiments, we additionally evaluated FairJudge on the external JudgeBench benchmark. Since JudgeBench is not built from the perturbation patterns used in our training pipeline, it provides a complementary test of unseen and more natural judge failures. FairJudge consistently outperforms the base model on both JudgeBench-Claude and JudgeBench-GPT-4o, suggesting that the gains of our method are not confined to the specific perturbation patterns seen during training, but extend to a broader and more general debiasing capability.
>
>
> ---
> ---
> ### **Response to W3**
>
> To further examine potential reward hacking, we measure the proportion of *tie* predictions before and after GRPO.
>
> | Model | JudgeBench | JudgeLM | PandaLM | SelfEval |
> |---|---:|---:|---:|---:|
> | Before GRPO | 0.0161 | 0.0802 | 0.1429 | 0.0456 |
> | After GRPO  | 0.0145 | 0.0770 | 0.1228 | 0.0481 |
>
> We observe that the tie rate decreases on three out of four benchmarks, with only a negligible increase on SelfEval. This suggests that GRPO does not lead to a degenerate strategy of predicting more ties. Instead, the consistency improvements are achieved without relying on tie inflation, indicating more reliable and meaningful alignment across evaluation modes.
>
> Moreover, consistency improvements are accompanied by stable or improved F1 performance across benchmarks, indicating that the gains are not achieved at the expense of judgment quality.
>
> This further suggests that the model is not exploiting superficial strategies, but learning more robust judgment behavior.
>
> ---
> ---
> ### **Response to W4**
>
> **We thank the reviewer for raising this insightful comment.** Our design is motivated by a mechanism-driven decomposition of the judging problem: SFT establishes a rubric-conditioned judging space, DPO reduces non-semantic biases, and GRPO enforces cross-mode consistency. These stages are not arbitrarily combined, but correspond to different sources of error in LLM-as-a-judge. Importantly, the staged design follows a curriculum-style training strategy, where the model first learns basic rubric-aware judgment, and is then progressively refined with more structured constraints. This design helps stabilize training and mitigate potential interference between objectives.
>
> **We sincerely thank the reviewer for your careful reading and thoughtful feedback.** We hope our clarification contributes to a clearer and more complete understanding of our work.

---

> > ### Author Rebuttal · Reviewer_65Ut · 2026-04-03
> >
> > Thanks for the detailed responses. My concerns are mostly addressed.

---

> > > ### Author Response · Authors · 2026-04-03
> > >
> > > We are pleased that our clarifications and additional experiments have addressed your primary concerns, and we sincerely appreciate your recognition of our efforts and the improvements made in this revision.
> > >
> > > We are also deeply grateful for your insightful and valuable comments, which have greatly contributed to strengthening the quality of our work.
> > >
> > > Thank you again for your time and support.

---

### Official Review · Reviewer_meQk · 2026-03-13

**Soundness:** 2
**Presentation:** 3
**Significance:** 3
**Originality:** 3
**Overall Recommendation:** 4
**Confidence:** 4

**Summary:**

This paper studies LLM-as-a-Judge and argues that a judge should be modeled as a conditional policy rather than a fixed scorer. The authors propose FairJudge, a framework that conditions the judgment on the input, rubric, and evaluation mode. They also build a dedicated training set and benchmark, and train the model in three stages to improve rule-following, reduce shallow biases, and increase consistency between pointwise and pairwise judgments. The experiments suggest that the method is competitive across several judge benchmarks, with the strongest gains appearing in consistency and practical efficiency.

**Compliance With Llm Reviewing Policy:**

Affirmed.

**Key Questions For Authors:**

Can you show a cleaner test for adaptivity, such as new rubrics or unseen evaluation settings? Right now, this claim is not fully supported.

How independent is FairJudge-Benchmark-1K from the training data pipeline? Please clarify this more clearly.

**Limitations:**

No. The paper should discuss limitations more directly,

**Strengths And Weaknesses:**

This paper studies an important problem in LLM-as-a-Judge. The authors try to improve judge models from three aspects: adaptivity, debiasing, and consistency. The overall framework is clear, and the training pipeline is well organized. The paper also shows decent results on several benchmarks. In particular, the gains on consistency and efficiency are meaningful. I think the topic is practical and relevant to the community.

However, the main weakness is that the paper claims more than it fully shows. The evidence for consistency is relatively clear, but the support for adaptivity and debiasing is still not strong enough. For example, I did not see a clean test under new rubrics or clearly different evaluation settings. So it is hard to know whether the model is truly adaptive, or just works well on similar data.

Another important weakness is the evaluation setting. The self-built benchmark seems to be close to the same data pipeline used in training. Because of this, the results may not fully show strong generalization. The paper would be more convincing if it had a more independent test set or stronger human evaluation.

Also, the paper does not clearly separate where the gains come from. The improvement may come from the policy view, the curated data, or the multi-stage training. The current ablation is not enough to tell which part is most important. So the scientific contribution is still somewhat unclear.

For presentation, the paper is generally readable and easy to follow. But the wording is a bit strong compared with the actual evidence. The paper would be better if the claims were more careful and more limited to what is directly supported.

---

> ### Author Rebuttal · Authors · 2026-03-31
>
> **We'd like to express our sincere gratitude for your careful readings and valuable comments.** We provide our feedback as follows:
>
>
> ### **Response to Weakness1**
>
> **Clarification on adaptive.**
>
> **We thank the reviewer for raising this valuable question.** To more directly evaluate adaptivity under **new evaluation standards (rubrics)**, we conduct additional experiments on **HelpSteer2**. This dataset is a multi-attribute human preference dataset, where each response is annotated along multiple dimensions (e.g., helpfulness, correctness, coherence), each corresponding to a distinct evaluation criterion (rubric). As such, it provides a natural multi-rubric evaluation setting, allowing us to test whether a model can adjust its judgments under varying criteria rather than relying on a single fixed standard.
>
> We measure alignment with each rubric using Kendall / Pearson / Spearman correlations. FairJudge consistently outperforms the base model across all three metrics (0.3677 vs. 0.3243, 0.4469 vs. 0.4082, 0.4157 vs. 0.3722), indicating better alignment under diverse evaluation criteria.
>
> Importantly, these dimensions correspond to **distinct and potentially conflicting rubrics** (e.g., conciseness vs. completeness), requiring the model to adjust its judgment criteria across settings. The consistent gains of FairJudge under this setup provide direct evidence that its behavior adapts to changing evaluation standards, supporting **truly adaptive** judgment rather than performance limited to similar data distributions.
>
>
> | Model | Kendall | Pearson | Spearman |
> |---|---:|---:|---:|
> | Base | 0.3243 | 0.4082 | 0.3722 |
> | FairJudge | **0.3726** | **0.4551** | **0.4214** |
>
> **Clarification on debiasing.**
> **We thank the reviewer for raising this valuable question.**  Due to the 5000-character limit, please refer to our response to **reviewer 65Ut (W2)** for detailed clarification.
>
> ---
> ---
>
> ### **Response to W2**
>
> **We thank the reviewer for raising this valuable question.** We conducted a hash-based overlap analysis between the provided training and test sets. We evaluated overlap under multiple exact matching definitions, including:
>
> - question-only
> - question + answer
> - question + answer + reference
> - pairwise (question + two answers)
> - pairwise (question + two answers + reference)
>
> After applying Unicode normalization, whitespace normalization, lowercasing, and additional punctuation removal, the overlap rate under all settings remains below **0.05%**.
>
> Therefore, we can confirm that there is no instance-level duplication or direct data leakage between the training and test sets.
>
> ---
> ---
>
> ### **Response to W3**
>
> **Thanks for the valuable comment**. Our intended contribution is an integrated framework rather than a full factorial decomposition. The policy-based formulation identifies three target judge properties; the corresponding data constructions instantiate supervision for them; and the curriculum-style SFT-DPO-GRPO pipeline is the concrete optimization mechanism. Our current ablations mainly isolate the training stages, showing these stage-specific signals are complementary and necessary, rather than fully disentangling modeling, data curation, and optimization as independent causal factors.
>
> ---
> ---
>
> ### **Response to W4**
>
> **We thank the reviewer for raising this important point**.  In the revision/rebuttal, we will refine several claims—particularly those regarding adaptivity, benchmark independence, and “structure matters more than scale”—to ensure they strictly align with the current evidence and do not go beyond what is supported by our experiments.
>
> **We sincerely thank the reviewer for your careful reading and thoughtful feedback.** We hope our clarification contributes to a clearer and more complete understanding of our work.

---

> > ### Author Rebuttal · Reviewer_meQk · 2026-04-04
> >
> > The rebuttal helps clarify the adaptivity claim, especially with the added HelpSteer2 results, and the overlap analysis also reduces some concern about direct leakage. However, I still find the benchmark independence and overall generalization evidence somewhat limited, and the contribution of each component remains not fully disentangled. So my concerns are only partially resolved.

---

> > > ### Author Response · Authors · 2026-04-05
> > >
> > > **Thank you again for the careful follow-up.** To address the remaining concerns as directly as possible, we added additional analyses below.
> > >
> > > ---
> > >
> > > **For component contributions (policy / data / training).**
> > >
> > > **Thank you for your insightful comment.** To clarify the intended role of the policy view, our policy formulation is not intended as a standalone module in a full factorial decomposition. Rather, it is the modeling perspective that specifies the target judge behaviors; the data construction instantiates supervision for these behaviors; and the staged SFT-DPO-GRPO pipeline provides the optimization mechanism. We will revise the paper to make this scope clearer.
> > >
> > > To more directly evaluate the contribution of data construction, we added a controlled SFT-only comparison: we fix the backbone and SFT objective and vary only the training data. Specifically, we compare SFT on the original raw data before our construction pipeline versus SFT on the data after our pipeline construction.
> > >
> > > **Metric notation.**
> > > PL / JL / FJ denote evaluation on PandaLM / JudgeLM / FairJudge-Benchmark-1K, respectively.
> > > A / P / R / F1 denote Agreement / Precision / Recall / F1.
> > >
> > > | Data setting | PL-A | PL-P | PL-R | PL-F1 | JL-A | JL-P | JL-R | JL-F1 | FJ-A | FJ-P | FJ-R | FJ-F1 |
> > > |---|---:|---:|---:|---:|---:|---:|---:|---:|---:|---:|---:|---:|
> > > | Base+SFT (before) | 73.54 | 65.46 | 64.30 | 64.67 | 75.17 | 61.36 | **61.99** | 61.43 | 66.59 | **64.11** | 55.88 | 57.45 |
> > > | Base+SFT (after) | **74.29** | **68.07** | **70.08** | **68.79** | **76.06** | **73.84** | 59.86 | **61.71** | **66.98** | 63.92 | **57.21** | **58.93** |
> > >
> > > This controlled comparison shows that the data after our pipeline construction consistently improves **Agreement** and **F1** across all three benchmarks. This provides more direct evidence that the data construction step itself is useful, rather than the gains coming only from later optimization.
> > >
> > > To then evaluate the contribution of training beyond data construction alone, we compare Base, Base+SFT, and the full FairJudge model:
> > >
> > > | Model | PL-A | PL-P | PL-R | PL-F1 | JL-A | JL-P | JL-R | JL-F1 | FJ-A | FJ-P | FJ-R | FJ-F1 |
> > > |---|---:|---:|---:|---:|---:|---:|---:|---:|---:|---:|---:|---:|
> > > | Base | 69.63 | 54.31 | 59.84 | 54.05 | 73.84 | 59.89 | 60.51 | 59.93 | 66.35 | 59.65 | 59.54 | 59.54 |
> > > | Base+SFT (after) | 74.29 | 68.07 | 70.08 | 68.79 | 76.06 | **73.84** | 59.86 | 61.71 | 66.98 | 63.92 | 57.21 | 58.93 |
> > > | Full FairJudge | **76.83** | **71.87** | **72.54** | **72.18** | **78.82** | 66.77 | **66.93** | **66.78** | **71.50** | **71.15** | **66.92** | **67.63** |
> > >
> > > Relative to SFT on the pipeline-constructed data, the full model yields further gains in **Agreement** and **F1** on all three benchmarks. In addition, in the original ablations, no reduced two-stage variant (SFT+GRPO, DPO+GRPO, or SFT+DPO) matches the full model. Taken together, these results support that both the data construction and the staged training pipeline contribute to the final gains. We will revise the wording to make clear that this is evidence for data and training contributions, rather than a full disentanglement of policy, data, and optimization as completely independent factors.
> > >
> > > ---
> > >
> > > **For broader generalization.**
> > >
> > > **Thanks for the valuable comment.**  To complement the external results already reported in the paper, we additionally evaluated on the external JudgeBench benchmark:
> > >
> > > | Model | JudgeBench-Claude | JudgeBench-GPT-4o |
> > > |---|---:|---:|
> > > | Qwen3-VL-8B | 0.5593 | 0.5943 |
> > > | FairJudge-8B | **0.6333** | **0.6600** |
> > >
> > > Since JudgeBench is external and not built from the perturbation patterns used in our internal pipeline, it serves as a complementary test of broader transfer. FairJudge-8B improves over Qwen3-VL-8B on both JudgeBench-Claude and JudgeBench-GPT-4o, suggesting that the gains are not confined to the specific internal perturbations used in our controlled experiments.
> > >
> > > ---
> > >
> > > **For training-set / benchmark independence.**
> > >
> > > **Thank you for raising this important comment.**  Regarding benchmark independence, our rebuttal already reported a hash-based overlap analysis under multiple exact-match definitions, with overlap below **0.05%**, which reduces concern about direct leakage. More broadly, the gains are not concentrated on FairJudge-Benchmark-1K alone. Against prior judge-oriented baselines (PandaLM-7B / JudgeLM-7B / FlexVL-7B), FairJudge-8B is also ahead on several external benchmarks reported in the paper, including PandaLM, JudgeLM, MLLM-as-a-Judge, and Reward-Bench. We therefore view FairJudge-Benchmark-1K primarily as an instance-disjoint but pipeline-aligned controlled benchmark for analyzing judge behavior, while broader transfer/generalization evidence is better reflected by the external evaluations. We will revise the paper to make this framing clearer.
> > >
> > > **We thank the reviewer again for the thoughtful follow-up and hope these additional analyses help clarify the remaining points.**

---

### Decision · Program_Chairs · 2026-04-30

**Decision:**

Accept (regular)

**Comment:**

The final reviewer scores for this paper are 4, 4, 4. Reviewers appreciated the policy-based formulation of LLM-as-a-Judge and the staged training pipeline targeting adaptivity, debiasing, and consistency, as well as the meaningful gains in consistency and efficiency across multiple benchmarks.

However, the main concern is that the paper’s claims are stronger than what is fully supported by the current evidence. In particular, while the results on consistency are relatively convincing, the support for adaptivity and debiasing remains limited, and the evaluation setting does not yet sufficiently establish generalization beyond the training pipeline. The authors are encouraged to clarify these points and moderate the claims in the camera-ready version.

Considering that all reviewers provided positive evaluations, I recommend acceptance.